# The Collection of Hyperspectral Measurements on Snow and Ice Covers in Polar Regions (SISpec 2.0)

Rosamaria Salvatori [1,†] , Roberto Salzano [2,*,†] , Mauro Valt [3], Riccardo Cerrato [2] and Stefano Ghergo [4]

1   CNR—Institute of Polar Sciences, 00010 Rome, Italy; rosamaria.salvatori@cnr.it
2   CNR—Institute of Atmospheric Pollution Research, 50019 Florence, Italy; riccardo.cerrato@iia.cnr.it
3   ARPAV—Avalanche Center, 32020 Belluno, Italy; mauro.valt@arpa.veneto.it
4   CNR—Institute of Water Research, 00010 Rome, Italy; stefano.ghergo@irsa.cnr.it
*   Correspondence: roberto.salzano@cnr.it
†   These authors contributed equally to this work.

**Abstract:** The data value of hyperspectral measurements on ice and snow cover is strongly impacted by the availability of data services, where spectral libraries are integrated to detailed descriptions of the observed surface cover. For snow and ice cover, we present an updated version of the Snow/Ice Spectral Archive (SISpec 2.0), which has been integrated into a web portal characterized by different functionalities. The adopted metadata scheme features basic geographic data, information about the acquisition setup, and parameters describing the different surface types. While the implementation of the IACS Classification of Seasonal Snow on the Ground is the core component for snow cover, ice cover is approached using different parameters associated with its surface roughness and location. The web portal is not only a visualization tool, but also supports interoperability functionalities, providing data in the NetCDF file format. The availability of these functionalities sets the foundation for sharing a novel platform with the community and is an interesting tool for calibrating and validating data and models.

**Keywords:** snow cover; ice cover; Arctic; Antarctic; spectral reflectance; hyperspectral data

## 1. Introduction

The cryosphere is a complex domain where surface snow and ice cover play a key role in the climate change framework. The spatial and temporal monitoring of these components is a critical task that requires both in-situ and remotely sensed observations. The description of snow and ice cover, defined as an aggregation of ice crystals with different sizes and shapes [1], can be approached using microphysical observations and optical measurements combining in-situ and remotely sensed data. Moreover, the spectral behavior of the surface is controlled by further factors associated with the microphysical conditions of the considered layer (snow grain shape and size), including the surface roughness and the chemical composition of the most superficial layer. The snow surface response in the visible wavelength domain is mainly affected by light-absorbing impurities (e.g., lithogenic dust, algae, soot) [2], whereas the snow properties in the short-wave infrared range (1400–2500 nm) are widely influenced by the size and shape of the ice crystals [3]. The spectral behavior in the visible range of compact ice and snow surfaces is similar, but in the short-wave infrared ranges, the former absorbs the incident radiation almost completely [1]. The calibration and validation of the satellite data of glacial and periglacial environments are therefore widely dependent on the knowledge of the optical behavior of snow and ice. From this point of view, the ground-based data obtained by field spectroradiometers represent an ideal and fundamental data source. Hyperspectral observations are an extremely interesting feature, especially when they are obtained during field surveys located in remote areas. The need for hyperspectral measurements on snow cover, obtained by different experimental setups [4], will be further evidenced by the deployment of hyperspectral sensors on satellite platforms

in the near future (e.g., PRISMA, CHIME, EnMAP). The combination of traditional manned observations [5,6] and autonomous acquisitions [7–11] will support such a demand, but the data value will only increase if observations are described in detail, especially regarding the acquisition setup [4], snow microphysics [12–15], and surface conditions [16].

The first pillar of the data value is data interoperability and, consequently, the data sharing of the described items using a standardized metadata profile. The availability of a snow-related data model [17], defined considering the first version of the Snow and Ice Spectral Library (SISpec 1.0) published by [18,19], matches this requirement. In fact, specific extensions have been combined in order to describe the three abovementioned information components (Base, Acquisition, and Domain). The paradigm shift from the crystal size-oriented classification to a more exhaustive classification that takes into account a complex mixture of crystal shapes and sizes is supported by the proposed scheme. The change of the data model allows the possibility of describing the snow surface as an ensemble of crystal types, sizes, shapes and their genesis, permitting the potential classification of the crystals into more than 30 classes, as indicated by the International Classification of Seasonal Snow on the Ground [20]. The snow crystallography is not the only feature described by the metadata profile, but it also allows a harmonization of microphysical properties of the surface (e.g., hardness and roughness), which were observed following the international guidelines. The encoding specifications drive the interoperability action through the preparation of NetCDF file formats compliant with INSPIRE and ISO guidelines. This background is a primary component for preparing a specific data service capable of showing, querying, and sharing its content with the community, or to the already available spectral archives (SA), spectral libraries (SL), and spectral information systems (SIS). The availability of the SISpec metadata profile also facilitates the connection between domain-specific collections and general-purpose systems (SA, SL, or SIS), improving machine-to-machine interactions with dedicated tools.

Following a survey about hyperspectral collections of snow surfaces, the available datasets could be grouped in domain-specific and general-purpose collections. The first group includes the SPECLIB SL [21], the ECOsystem Spaceborne Thermal Radiometer Experiment on Space Station (ECOSTRESS) SL [22], the SPECCHIO SIS [23], and the LUCAS database with its SL [24]. The second group includes two soil-oriented collections, namely, the INTA SL [25] and the Global vis-NIR SL [26], and one snow-related SISpec SL [18]. The interoperability issue is, of course, a major aspect, since some collections, like ECOSTRESS and SPECLIB, consist of text files reporting the single measurements (spectra) associated with a limited amount (about 20) of unstandardized metadata. Nevertheless, SPECCHIO and INTA are characterized by a more detailed and standardized data model with more than 50 attributes. The Global vis-NIR SL and LUCAS SL are dedicated to soils, and their structure has specific attributes (8 and more than 30, respectively) for chemistry and physics that are not suitable for hyperspectral measurements on snow surfaces. Regarding snow observation databases, ECOSTRESS, SPECLIB, INTA, and SPECCHIO provide few spectra with a very coarse description, which can be difficult to utilize in certain studies. SISpec 1.0 [18] is the only fully snow-oriented collection where snow measurements are fully described in compliance with the international classification [20]. This library was released with physical support and this limitation required an upgrade to SISpec 2.0, with the creation of a web service that must be user-friendly, flexible, and interoperable. The aim of this paper is to present such a web service and associated database, which can promote the sharing of available measurements and the harvesting of novel observations in snow-covered areas.

## 2. Methods

The presented spectral library contains data collected during field campaigns in polar regions between 1998 and 2011. The ability to analyze flat surfaces is a prerogative that makes polar areas the most suitable for collecting snow and ice spectral measurements, because the spectral response of these surfaces is not appreciably affected by the presence

of nearby reliefs and vegetation or by light-absorbing impurities of anthropogenic origin. The field data were collected during cold and clear sky days (with air temperatures always below 0 °C), to avoid bias due to the snow cover melting. All of the selected sites were located in wide and plain areas in order to minimize slope and adjacent pixel effects, and to be clearly identifiable at Landsat satellite data resolution [18]. The collection includes more than 250 spectra obtained using a standard procedure both for the acquisition of reflectance measurements and for the manual snow observation [20].

### 2.1. Regions of Interest

The dataset includes 152 hyperspectral measurements acquired in the Arctic, where radiometric and snow/ice data were collected from 26 different measurement sites located in the Brøggerhalvøya (Brøgger Peninsula), up to 160 km from the International Research Station of Ny-Ålesund (Figure 1a). The field campaigns were carried out in six spring seasons (1998, 2000, 2001, 2003, 2010, and 2011). The selected region is characterized by flat snow/ice surfaces (mainly tundra snow and various ice covers), large enough to be detected on satellite images and far enough from dense human settlements, in order to provide relatively uncontaminated snow spectral signatures. The peninsula is characterized by a relatively large variability of snow conditions in the framework of relatively short distances due to the different exposure of the coastline towards local-scale meteorology [27,28].

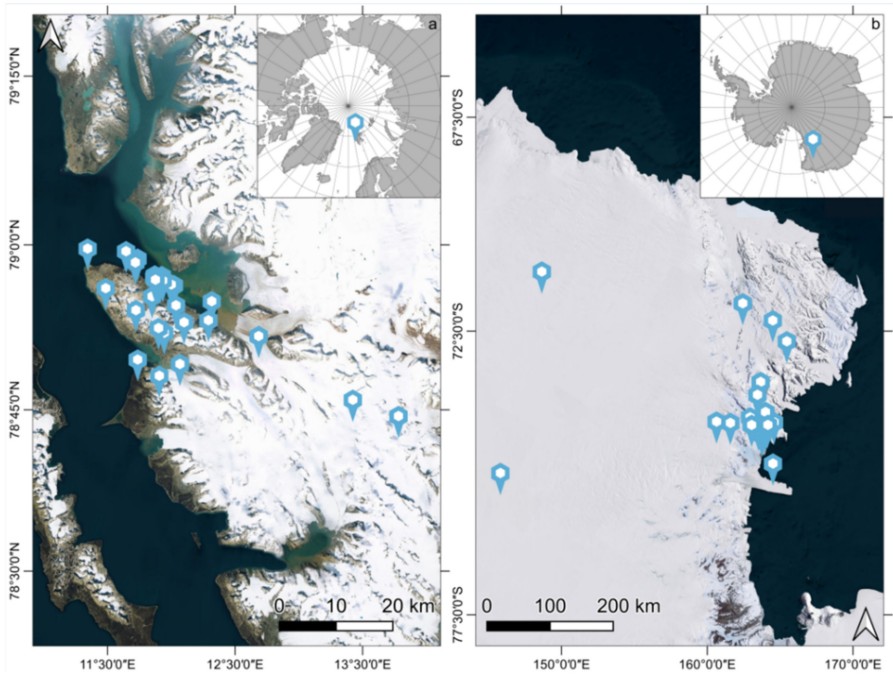

**Figure 1.** Localization of the available measurement sites in the Arctic region (**a**) and in the Antarctic continent (**b**).

The 105 spectral snow reflectance observations in Antarctica were acquired from 25 measure sites at Terra Nova Bay (Victoria Land—Ross Sea Region) in the Austral summers of 1996, 1998, and 2002. The location of the investigated sites is shown in Figure 1b. A large variety of glacial landforms and snow/ice types were detected due to the intense glacial activity and to the major difference in elevation between areas next to the coast and areas on the large inner plateau.

### 2.2. Hyperspectral Measurements

The snow and ice hyperspectral measurements were acquired by different field spectroradiometers: Fieldspec FR and FR3 (Analytical Spectral Devices Inc., Boulder, CO, USA), covering the wavelength range between 350 and 2500 nm. Measurements were acquired

using a hemispherical–conical setup (Figure 2), where the calculated reflectance factor was estimated as the ratio of incident solar radiation reflected from the snow target and the incident radiation reflected from a white reference Spectralon (about 30 cm × 30 cm), known as a Lambertian reflector. The acquisition protocol was standardized, with the sensor optic fiber on a tripod 50 cm above the surface target in a nadir position. The radiometer was used with the bare optics that correspond to a field of view of 25°, thus covering ground areas of about 23 cm in diameter. All of the measurements were acquired by positioning the target towards the sun and minimizing interferences on the surface from the operator and the tripod. The absolute reflectance was obtained by multiplying the reflectance factor by the calibration coefficient of the reference panel. Possible sources of errors or noise in field spectroradiometry could be operational due to the incorrect viewing geometry during the data acquisition, or due to internal (e.g., random noise produced by electronic components of the instrument) or external (e.g., the atmospheric water vapor absorption, the low atmospheric irradiance at wavelengths of 1400 nm and beyond 1700 nm) factors causing a low signal-to-noise (S/N) ratio. To avoid incorrect or anomalous reflectance values and/or patterns, especially in the visible and infrared wavelengths, a correct orientation of the spectroradiometer over the calibration panel and the snow surface target is necessary. The optimization of the S/N ratio is a major issue in field measurements; the selected compromise was found using a sample average from 10 to 50 acquisitions. For every target, a statistically meaningful sample of the surface was obtained by acquiring 20 to 30 spectral curves; the evaluation of the signal's stability and of the surface heterogeneity contribute to defining the final number of acquisitions. This procedure increases the spectral characterization of the snow target—the higher the number of spectral curves, the smaller the random errors. No spectral polishing algorithm was applied to the published data. Values in the wavelength range around 1400, 1940, and 2400 nm are affected by the presence of atmospheric water vapor [29].

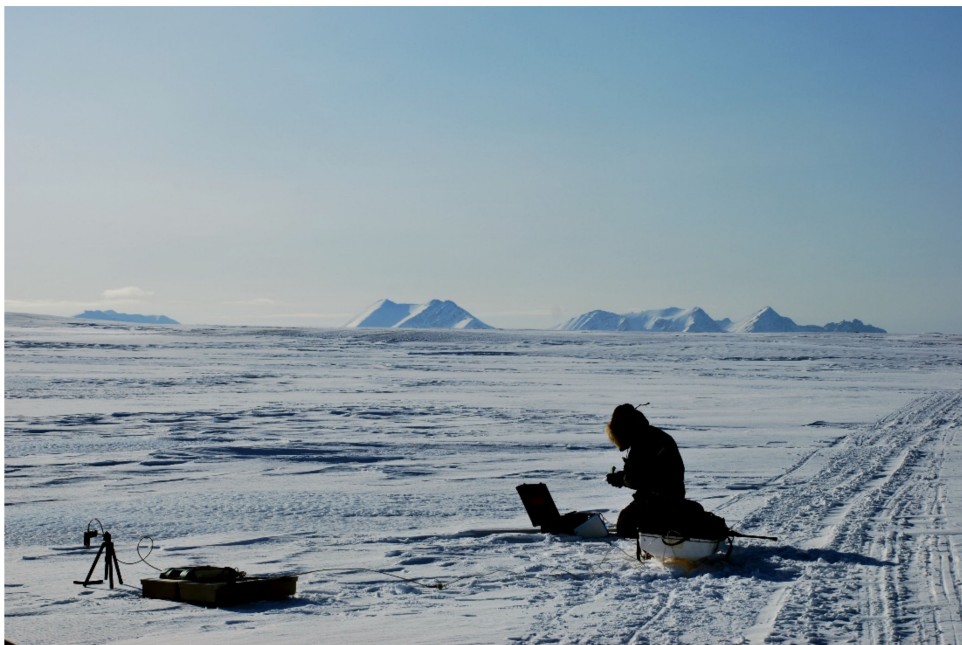

**Figure 2.** Field hyperspectral observation setup.

### 2.3. Surface Characterization

The description of each snow/ice target was performed following a protocol based on the preliminary identification of the most representative target of the entire area (about 100 × 100 m) in terms of surface type: ice and snow cover. The surface roughness is a major feature in both cases, and the selected protocol for such a description was based on acquiring a visual overview of the surface characteristics with a terrestrial image as well

as to measure the roughness in terms of the length and height of the identified geometric elements (e.g., ripples, furrows). The different pattern and size of the surface elements may affect the spectral response due to shadowing effects and backscattering. Surface roughness (furrow distance and depth) was also measured in mm, in accordance with [20].

### 2.3.1. Snow Cover Microphysics

The description of the microphysical characteristics of the snow was carried out by determining the shape and size of the snow grains based on the observation of the individual grains, with a magnifying glass, on a graduated crystals card, as indicated by [20]. Considering that the surface of the snow is often made up of a mixture of grains of different sizes and shapes, the three most abundant grain types were reported for each target surface. Snow observations were performed on the selected target area immediately after the spectral measurement. A conventional survey of the first 20 cm depth of the snow surfaces was carried out at all measurement sites in order to measure the hardness, density, and liquid water content of the snow cover. Hardness measurements were performed using the hand test [30]; the snow density of the first 10 cm (from the surface) was measured using a core drill with a volume of $10^{-4}$ m$^3$ and a properly calibrated steelyard scale. The liquid water content was estimated with a SnowFork (Toikka Engineering Ltd., Espoo, Finland). The temperature of the air and snow (10 cm below the surface) was measured at each site immediately after the spectra were acquired. Snow and air temperature measurements were carried out using a digital contact thermometer.

### 2.3.2. Ice Cover Classification

The description of the ice cover followed the principles defined in [31], where the major cryospheric components that complete the framework with the snow cover are glaciers and ice caps, freshwater ice, and sea ice. The ice cover classification in SISpec is limited to the geographical framework of the considered site: sea ice, glacier ice (the surface of the glacier not covered by snow), and lake or river ice. According to the International Classification of Seasonal Snow on the Ground [20], codes with two uppercase letters are selected for these iced surfaces (IS) and two lowercase letters: we use ISsi for sea ice, ISgl for glacier ice, and ISlr for lake or river ice.

### 2.4. Other Info

Environmental parameters (e.g., sky conditions, air temperature, and humidity), the GPS coordinates, the elevation of the observation site, and the date and time of the spectral measurement were also acquired for each measurement site. The date and time data were then used to calculate the elevation and azimuth of the sun. When possible, pictures of the target and panoramic photos of the measuring site were acquired.

### 2.5. SISpec Spectral Information System

The SISpec database and management procedures (Figure 3) are designed to facilitate the storage, organization, and elaboration of the spectral data and of the supplementary observations collected during field measurements. The SISpec web service (https://niveos.cnr.it/SISpec, accessed on 2 May 2022) supports specific tasks that steer the implementation of the data management/data output tool system, enhancing the following actions: (i) allows local operators and visiting users constant access to basic information and processed data; (ii) centralizes data management activities; and (iii) standardizes the most-used visualization and processing procedures. The relational database management system (RDBMS) used for this project is MySQL (https://www.mysql.com/ accessed on 2 May 2022) and the server-side procedures are written using PHP (https://www.php.net accessed on 2 May 2022) as a programming language. To facilitate client-side interactions with users, the procedures contain code in JavaScript. The relational structure of the database was designed considering the SISpec web system purposes and the different types of data collected during the field campaigns. The SISpec core component (left side of Figure 3)

supports the hyperspectral data (reflectance values as well as wavelength ranges) with the information about the observation site (metadata about the geographic location), the primary surface characterization (description and metrics), and the environmental conditions (using categories and numerical variables) during the acquisition of the data and photographic material regarding the target. Additional information is present in the nivological component (right side of Figure 3) for the snow cover, and the attributes are organized following the international convention available in the literature [20]. The most relevant information of the nivological component is stored as coded fields that need support tables to supply a more user-friendly appearance for the output forms. The code conventions used in these support tables are reported in [20]. Reflectance spectra in the database are stored as pointers to external files as well as the nivological profiles, the target photos, and the metainformation (in NetCDF format) files. The whole system is supported by predefined standardized queries, which facilitate the navigation and visualization of the available dataset. These functionalities are based on code lists developed using specific domain knowledge, such as the snow classification, surface classification, satellite band specifications, and geographic toponyms. The SISpec web system is equipped with an authentication procedure currently only used for database table maintenance, but this can be easily upgraded for the implementation of new features of the system such as new contributions from other users or the direct download of selected data.

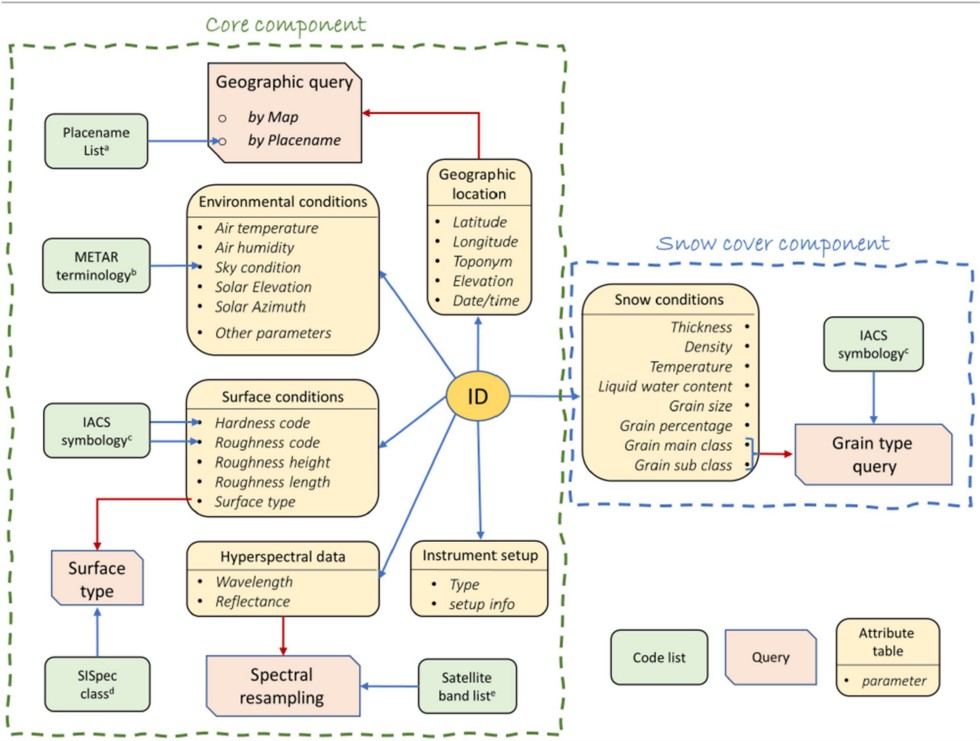

**Figure 3.** Database structure and relational tables breakdown: the core and nivological components are characterized by attribute tables, functionalities, and code lists. [a] is the placename list defined in [32], [b] refers to the METAR terminology reported in [20], [c] refers to the IACS classification [20], [d] are the classes defined in this paper combining [20,31], [e] are the bands declared by each satellite platform [33–36].

## 3. Results

The SISpec collection presented in this study includes the dataset published in the first version [18] and has been updated with newer observations acquired in recent years. While the accessibility to the collection is approached by developing a specific web service, the interoperability is handled by preparing data following the SISpec metadata profile defined by [17].

### 3.1. SISpec Web Portal

The published web service is based on a user interface that supports different functionalities, aimed at data querying and visualization. The collection can be queried using two different strategies (Figure 4) based on geographic location and snow microphysics, respectively.

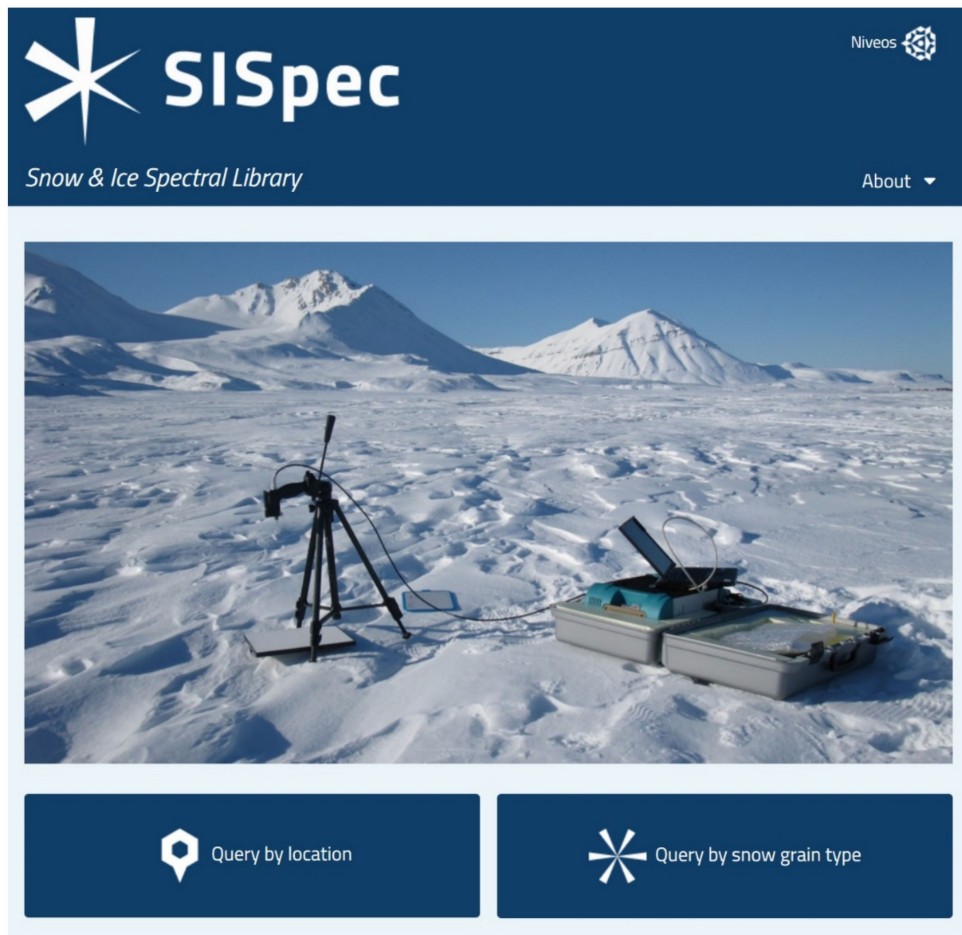

**Figure 4.** The search features of the SISpec web portal.

The geographic search is currently limited to the two regions of interest (Svalbard and Antarctica) by constraining the searchable geographic extent. The search functionality focusing on snow microphysics is currently limited to the snow grain type, defined by [20]. Both search queries produce a standardized output (Figure 5), which supplies the list of all records satisfying the search parameters.

The left panel of the search output is the summary of the results obtained, reporting the file ID, the geographic location, the acquisition time, the grain type symbology, and the snow layer thickness. Each record can be further displayed interactively with a single or multiple selection. The full data information can only be displayed for each measurement, and the right-side panel reports the complete metadata description of the selected observation in terms of geographic location, nivological properties, environmental conditions, and acquisition setup. The output list of a query provides the functionality to plot each of the reported records with a specific panel displaying the reflectance spectrum (Figure 6).

Such a representation also supports the multiple selection of up to six spectra, which can be displayed as separate curves in a collated plot, as an averaged spectrum, or as a plot grid with single records. The proposed web interface is therefore aimed at supporting the analysis and interpretation of satellite images. This final objective implies the availability of a specific tool focused on resampling selected spectra in the corresponding intervals of the bands of different satellite sensors. It is possible to visualize the curves not only in

the satellite bands that have the same spatial resolution, but also to select the option that allows the visualization of the resampled value in all of the available bands. This latter opportunity allows users to support many different applications to study snow cover. In addition, for the multiple selection option, the satellite bands resampled output is available (Figure 7). The sensor portfolio included in the presented tool includes the band list of instruments deployed on different platforms: Landsat 7 [33]; Landsat 8 [34]; Sentinel-2 [35]; and Terra/Aqua [36]. The band resampling was developed by calculating the resampled values using the specific full width half maximum (FWHM) of each sensor band.

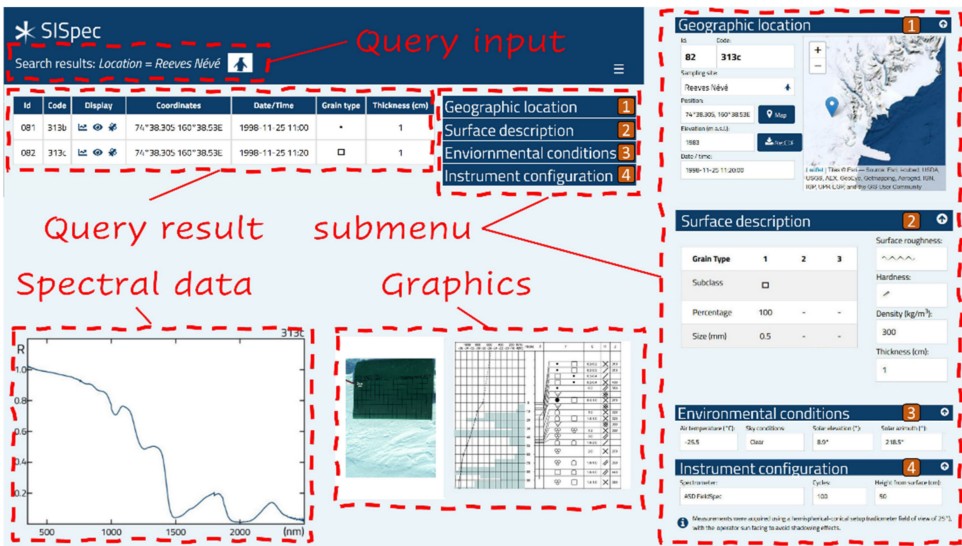

**Figure 5.** Results of the query by location and available information of the Reeves Névé site. Information sections rearranged for clarity.

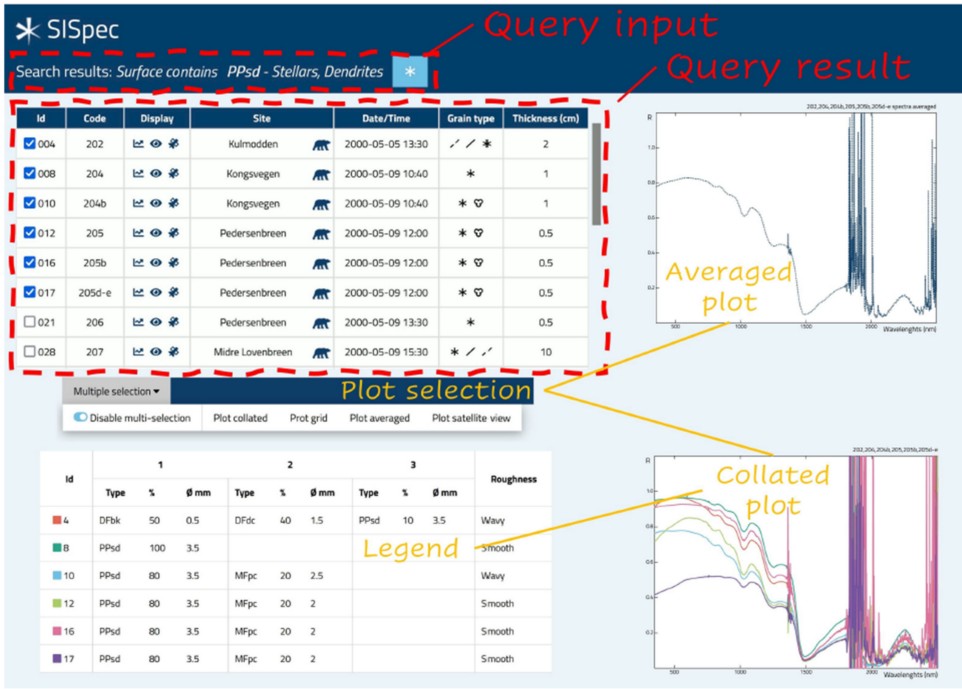

**Figure 6.** Results of the query by grain type PPsd with spectrum visualization and multiple selection. Information sections rearranged for clarity.

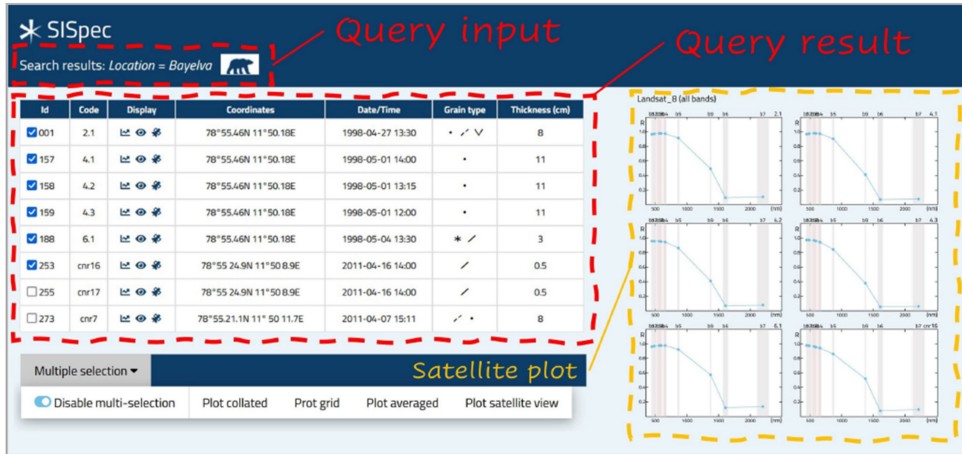

**Figure 7.** Results of the query by location and spectrum for the Bayelva sites. Example of band resampling considering the sensors deployed on the Landsat 8 platform. Information sections rearranged for clarity.

### 3.2. SISpec Interoperability

The interoperability issue of the data value chain was approached considering the latest standards and the available convention to prepare sharable and reusable datasets. The preparation of datasets in the NetCDF file format, following the encoding technical specification prepared by [17], provided a solution for having a formal and shared standardization aimed at producing well-documented and sound metadata for hyperspectral measurements of the snow/ice surfaces. The dataset was prepared with single files for each measurement that can be downloaded using the dedicated functionalities. The prepared data are fully described following the ISO and INSPIRE guidelines. They are ready for machine-to-machine interactions and can be easily read by different open tools. Metainformation and data for each measurement, in NetCDF format, are downloadable through the field acquisition form, which provides a map of the sampling site.

## 4. Discussion

### 4.1. Usefulness of the SISpec Archive

The available records presented in the SISpec web portal include observations acquired both in the Arctic and Antarctic regions, 152 and 105, respectively. The archive provides hyperspectral measurements obtained on snow (209) and ice (48) covers, which are described in detail from the acquisition point of view, as well as in terms of surface and nivological conditions. The observations are representative of a large variety of environmental conditions, considering that the air temperatures covered a range from $-21$ to $3\,^\circ$C in Svalbard, and from $-34$ to $-2\,^\circ$C in Antarctica. The study sites included 66% locations in coastal areas (i.e., below 100 m a.s.l.), 17% low elevated lands between 100 and 500 m a.s.l., 6% medium elevated areas between 500 and 1000 m a.s.l., and 11% high elevated areas above 1000 m a.s.l. The sky condition is additional metadata information that has a double impact on both the snow metamorphism and the reflectance estimation. The primary target of the considered field campaigns was to maximize measurements obtained under clear or white sky conditions with 47% and 20% of the available records, respectively. The key parameter for discriminating between ice and snow cover was the hardness, but the description of snow surfaces was completed by snow densities (with an average of $253 \pm 107\,\mathrm{kg\,m^{-3}}$) and a roughness of almost smooth in more than 50% of observations (Figure 8). Furthermore, measurements showed conditions where the average snow temperature in Svalbard was $-6.2 \pm 4.3\,^\circ$C and in Antarctica was $-14.6 \pm 8.7\,^\circ$C. Therefore, the snow surfaces were described using the IACS Classification of Seasonal Snow on the Ground, and interesting features were obtained about the snow microphysics. The observed surfaces were, in fact, characterized by single shape classes in 33% of cases, by two

different shape types in 51% of occurrences, and by three components in the rest of the available measurements. The median size of the observed snow grains varied from recently formed crystals (PP, DF, and SH) at 1.2 mm, to surface modified forms (FC and RG) at 0.4 mm, and to melt forms (MF) at 1.65 mm.

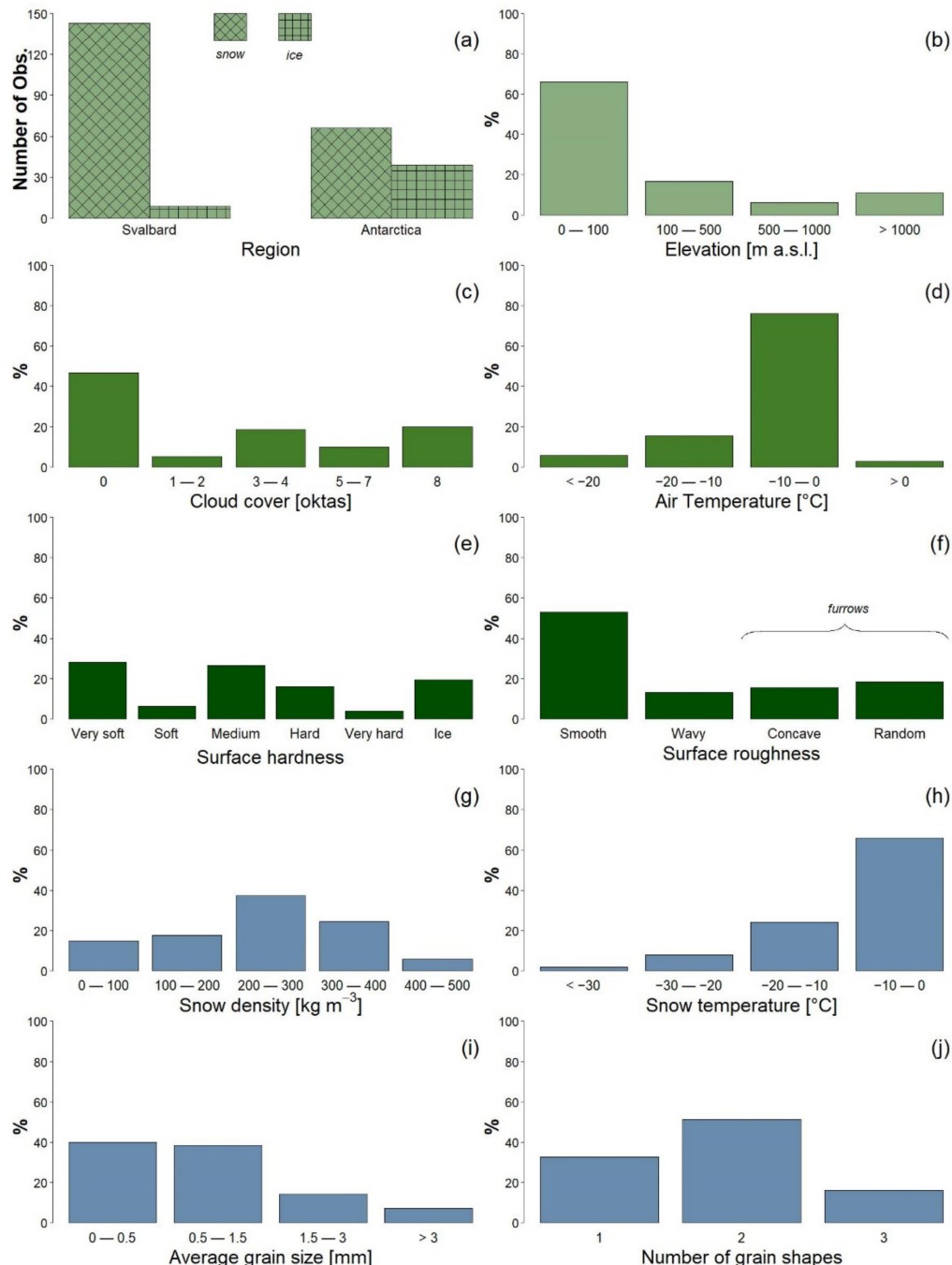

**Figure 8.** Summary plot of observations: (**a**) region of interest, (**b**) site elevation, (**c**) cloud cover during the acquisition, (**d**) air temperature, (**e**) surface hardness, (**f**) surface roughness, (**g**) snow density, (**h**) snow temperature, (**i**) average snow grain size, and (**j**) number of identified snow crystal forms.

All of these data are useful metrics that support the processing of hyperspectral measurements (Figure 9). The description of snow and ice cover is a major scope of the support provided by the SISpec web portal and it can be approached considering preliminary criteria aimed at discriminating different surface types. The SISpec archive provides continuous variability between different surface types that are significantly discriminated between

snow (80%)- and ice (20%)-covered end members. The separation between these surface types in terms of reflectance increases from the visible range to the short-wave infrared wavelength domain (above 1100 nm). This behavior is, of course, consistent with the literature evidence [1], but the management of different metadata patterns associated with the two different surface types is a primary requirement. The analysis of snow surfaces considering the IACS classification [20] is a major challenge of the SISpec web portal, since it is possible to combine and share a range of information impacting on the final spectral behavior of the observed surface. The potentialities offered by the web portal include the sharing and visualization of data, described as complex mixtures of different snow grain shapes, that will enhance studies concerning radiative transfer models [2,12,15] and novel sensing capabilities of the snow cover [37–39].

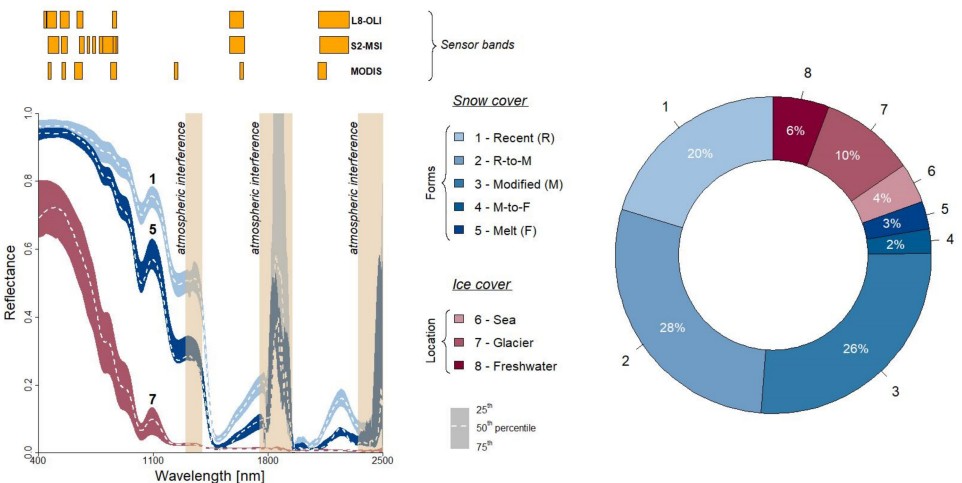

**Figure 9.** Aggregation of hyperspectral measurements of snow and ice cover (**left**) and statistics on the occurrence of different snow forms and ice locations (**right**). Snow cover is aggregated considering the occurrence of different snow grain types.

The proposed approach is based on aggregating the different snow forms, considering the texture modification from recently deposited or precipitated crystals to advanced melting and refreezing of forms. While precipitated particles (PP), decomposed and fragmented grains (DF), and surface hoars (SH) are included in the recent forms, rounded grains (RG) and faceted crystals (FC) occur in the modified forms, produced by both equilibrium and kinetic growth metamorphism. Furthermore, the modified forms are dominated by equilibrium gradient grains (RG), since the environmental conditions observed during the fieldwork favor the occurrence of those forms on the surface. The occurrence of melt forms (MF) represents the final stage before the transformation in perpetual snow and ice formation. It is, of course, not possible to have pure terms, and mixed elements must be considered. Five classes are presented from recently formed snow grains, to modified forms, and finally to refrozen melt snow, where the spectral features are gradually lower in the whole spectrum range. Furthermore, the decrease is more intense in the short-wave infrared, as expected from the lengthening of the mean free path through the ice in relation to the decrease of the specific surface area already observed by [3]. The analysis of iced surfaces is, on the other hand, not mature enough, since the number of available observations is limited. From this perspective, the SISpec web portal only supports a geographic classification based on discriminating ice formations (IF) in the snowpack from iced surfaces associated with sea ice (ISsi), with freshwater entities such as lake and rivers (ISlr) and glacier areas (ISgl).

The flexibility of the SISpec archive is based on the ability to describe complex mixtures of several snow grain types instead of using sharp separation between single elements considering just size, shape, or density. The detailed description offered by the IACS classification [20] is, in fact, an added value that must be considered during the intercomparison

between ground-based and satellite observations. It is, therefore, limiting to discriminate between wet or dry snow cover, coarse or small-sized snow crystals, and new or old snow. SISpec allows a full description of complex surface layers, where different and heterogeneous snow grains could represent the deposition of new fallen snow onto intensively melted snow cover types. The opportunity to describe mixed surfaces where the snow cover is momentarily deposited on ice covers is also worthy of note. Finally, the aim of the SISpec functionalities is to support the full description offered by the IACS Classification of Seasonal Snow on the Ground, and to match the requirements of hyperspectral measurements from different disciplines and communities.

*4.2. SISpec Collection in the Spectral Archive Framework*

The integration of the international classification of snow [20] into a web service represents a novel tool for visualizing and sharing spectroradiometric observations of snow and ice surfaces. The available services, such as the ECOSTRESS SL, include snow measurements in the water type, distinguishing between granular snow (coarse, medium, and fine) and frost. The size distribution of the snow cover is, unfortunately, a limiting approach since the grain shape and composition, as well as the surface roughness, are driving factors of the final spectral behavior of the surface covered by snow and ice. This is the reason why the key novel aspect of the SISpec web portal is the description of several crystal types, limited to three different classes, where dimensions and crystal genesis define more than 30 different classes. The description of additional microphysical properties (hardness and roughness, for example), estimated with internationally recognized methods, is an additional key feature of the presented data portal. Having a fully described dataset is crucial in terms of gathering information on the acquisition setup, the surface condition, and the microphysical framework. It is a powerful tool featuring further advances on interpreting the optical behavior of snow and ice cover.

The flexibility of the SISpec data service supports different interactions between the offered dataset of the available snow/ice standard classifications. While the IACS classification supports avalanche-oriented communities, the available metadata and data format include the ability, among others, to revisit the snow classification using different perspectives: e.g., microphysical or meteoclimatic. The integration of the web portal to the NetCDF file format represents an ideal solution for human and machine-based interactions. The proposed strategy increases the data value of the SISpec dataset, offering the perspective to facilitate the interaction between smaller databases and wider services useful for different communities. The integration of the SISpec data model into collections of spectral data supported by other databases will be possible, developing a middleware aimed at translating the SISpec schema into selected wide-audience services. The opposite flow direction is unfortunately limited by the occurrence of snow measurements in other services (SPECLIB, ECOSTRESS, and SPECCHIO). While it is possible to ensure a complete overlap between the considered collections and SISpec, the opposite direction is limited by the available metadata components. The match between different attributes included in the SISpec scheme and the other collections is the only requirement for the translation tool. On the opposite side, the to-SISpec process is more critical due to missing snow information, and gaps in terms of mandatory SISpec attributes. The SISpec web portal represents, in conclusion, an ideal and flexible solution for the cryosphere community to share data, and the web portal is potentially open to include datasets acquired from different locations, being compliant with FAIR principles and INSPIRE guidelines.

**5. Conclusions**

The presented version of the Snow/Ice Spectral Archive (SISpec 2.0) is an innovative web portal where hyperspectral measurements on ice and snow cover are available in terms of accessibility and interoperability. The revisited spectral library is now integrated with detailed descriptions of the observed surfaces, and the adopted metadata scheme includes information about basic geographic features, the acquisition setup, and most of all, about

the parameters describing the different surface types. While snow cover is fully described following the IACS Classification of Seasonal Snow on the Ground, the approach to ice cover involves applying different parameters regarding surface roughness and location. The web portal is designed as a visualization tool, but it also supports different interoperability functionalities. The use of a standardized NetCDF file format allows the presented archive to connect to other platforms and to prepare the ground for investigating the relationship between the optical classification of the surface and the macro-/microphysical conditions of the observed cover. The availability of these functionalities sets the stage for sharing a novel platform with the community, creating an interesting tool for calibrating and validating data and models.

**Author Contributions:** All authors are intellectually responsible for the conducted research, work design, and manuscript preparation. Conceptualization, R.S. (Rosamaria Salvatori), S.G. and R.S. (Roberto Salzano); methodology, R.S. (Rosamaria Salvatori), M.V. and S.G.; software, R.S. (Rosamaria Salvatori) and S.G.; validation, R.S. (Rosamaria Salvatori), R.S. (Roberto Salzano) and M.V.; data curation, R.S. (Rosamaria Salvatori), M.V. and R.S. (Roberto Salzano); writing—original draft preparation, R.S. (Rosamaria Salvatori) and R.S. (Roberto Salzano); writing—review and editing, R.S. (Rosamaria Salvatori), R.S. (Roberto Salzano), R.C., M.V. and S.G.; visualization, R.S. (Rosamaria Salvatori), R.S. (Roberto Salzano) and R.C. All authors have read and agreed to the published version of the manuscript.

**Funding:** The development of the SISpec web portal is supported by the CRASI project (PNRA18-00131) and by the EcoClimate project (PRA21-19) in the framework of the reusing strategy of past datasets and the data management and harmonization of future novel datasets.

**Data Availability Statement:** The dataset is available for research purposes directly on the website after user registration.

**Acknowledgments:** The authors are extremely grateful to the staff of the ARPAV-Avalanche Center of Arabba (Renato Zasso, Anselmo Cagnati) and the CNR staff involved in the scientific support (Ruggero Casacchia) and the outstanding logistical support (Roberto Sparapani, Paolo Plini, Alessandro Mei) received during the field campaigns. We would like to acknowledge Giorgia Ghergo for the graphic project of the website and Massimiliano Olivieri for the server configuration.

**Conflicts of Interest:** The authors declare no conflict of interest.

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
