# Peer review of "The Collection of Hyperspectral Measurements on Snow and Ice Covers in Polar Regions (SISpec 2.0)"

_remotesensing, doi:10.3390/rs14092213_

Round 1
Reviewer 1 Report
See my comments in the attached pdf file.

Author Response
We thank the reviewer for his valuable comments. Here below we report the comment and our response.
Page 1, lines 26-27: "The monitoring of these components in terms of extension during time is a critical task that requires both field and remote sensing observations. " Awkward. Please rephrase. I would rather say: "The spatial and temporal monitoring of these components is a critical task that requires both field and remote sensing observations."
We accepted the suggestion and rephrased the sentence.
Page 1, lines 47-48: grammatical error "the data value will increase only if observations will be described in detail" -> "the data value will increase only if observations are described in detail".
We corrected the typo as suggested.
Page 2, line 51: "metadata profile". You could also use the more usual expression "data model".
We replaced as suggested “data model” instead of “metadata profile” when the intention is to refer with a more general concept. The original “metadata profile” is more specific and possibly more appropriate in some sentences.
Page 2, lines 88-89: "This library was released on physical support and this feature is the major gap that required the upgrade to SISpec 2.0 ". I would rather say : "This library was released on physical support and this limitation required the upgrade to SISpec 2.0".
We modified the text as suggested.
Page 2, line 90-91: "The aim of this paper is to present such a collector, which can promote the sharing ". "Collector" is not an appropriate word in that context. I would propose: "The aim of this paper is to present such a web service and associated database, which can promote the sharing."
We modified the text as suggested.
Page 4, lines 164: "acquiring a graphic overview of the surface characteristics". I would rather say : "acquiring a visual overview of the surface characteristics".
We modified the text as suggested.
Figures 3, 5, 6, and 7: the quality of these figures is bad and thus they are difficult to read. Please increase the resolution in dpi.
The figures have been updated with a higher resolution version.
Page 9, "Metainformation, in NetCDF format, is downloadable through the field acquisition form which provides a map of the sampling site." Not clear to me to which web page you are referring to and which data can be retrieved exactly. Please provide a screen shot of the page and indicate how the data can be retrieved in netCDF format.
We updated the Fig. 5 with the requested information. The web service portal has been updated as well but we are still solving some GDPR issues in order to be fully automated for downloading each netcdf file .
Page 9, line 321: it seems strange to have a mean snow temperature lower in Svalbard than in Antarctica! Please give an explanation.
We thank the reviewer for reporting this typo. We checked and corrected them, now the sentence is: “Furthermore, measurements showed conditions where the average snow temperature in Svalbard was -6.2 ± 4.3 °C and in Antarctica was -14.6 ± 8.7 °C”.
Page 11, line 367: "as expected by the decrease of the specific surface area". I would add "as expected by the lengthening of mean free path through the ice in relation to the decrease of the specific surface area" to state clearly the link between the optical and textural properties of snow.
We modified the text as suggested.
Page 12, line 407-408: "an ideal solution for human and machine-based interactions. " Will you provide an API for automatic request and retrieval of data from the SISpec ? SISpec Web service, Nivological data. For an easier identification of the grain subclasses,
The download procedure is next to be automated with a registration step and we are solving some GDPR issues. The availability of APIs will be a future developpment since the machine-to-machine interaction is an already obtained result.
I would suggest to add in the pop-up window associated to each symbol, a short description in addition to the 4 letter code (ex. FCxr: Rounding faceted particles)
The web service portal has been modified as suggested.
Reviewer 2 Report
The scientific research which presented in the manuscript "The collection of hyperspectral measurements on snow and ice covers in polar regions (SISpec 2.0)" is dedicated to discribing of the specific database on snow and ice. It includes the measurements in the different spectral diapazone of the snow over twenty centimeters in depth. Authors suggested improved model of of the snow, which also gives detailed classification of different snow types. So, this research is very useful for the wide round of glaciologists.
I found this scientific research is very original, valuable and actual for the specialists in glaciology. The database in improve our knowledge about snow and ice. It includes 250 spectra which collected very varied environmental conditions for the number of field seasons in two poles: Svalbard and Antarctica. As I know, this kind of research it was not published before. I find the methodology of the authors are very good and I can suggest only concentrate to inprove the general quality of measurements and WEB service.
The main conclusions consistent with the evidence and arguments presented in the manuscript and confirms the scientific valuable of the database and the new improoved model for the glaciological research and I also think for more wide round of mathematical modelling.
The manuscript consists of twenty-six as recent as relative old scientific publications which is also good review of the current research in this scientific brunch. The important aspect is the visual presentation of the scientific research. It is also important as and scientific description. Well illustration helps reader to better understanding of the scientific research.
It necessary to say, the quality of the figures of the manuscript are very good and it also the merit of the authors.
Author Response
We thank the reviewer for his valuable comments.
Reviewer 3 Report
The quality of the article is very good and the results are well presented and sound.
Author Response

(The authors gave the same response as above.)
